# Development of Ductile and Durable High Strength Concrete (HSC) through Interactive Incorporation of Coir Waste and Silica Fume

**DOI:** 10.3390/ma15072616

**Published:** 2022-04-02

**Authors:** Babar Ali, Muhammad Fahad, Shahid Ullah, Hawreen Ahmed, Rayed Alyousef, Ahmed Deifalla

**Affiliations:** 1Department of Civil Engineering, COMSATS University Islamabad-Sahiwal Campus, Sahiwal 57000, Pakistan; 2Department of Civil Engineering, University of Engineering & Technology (UET), Peshawar 25000, Pakistan; fahadkhan@uetpeshawar.edu.pk; 3Earthquake Engineering Center, Department of Civil Engineering, University of Engineering & Technology (UET), Peshawar 25000, Pakistan; shahid.ullah@uetpeshawar.edu.pk; 4Department of Highway and Bridge Engineering, Technical Engineering College, Erbil Polytechnic University, Erbil 44001, Iraq; 5Department of Civil Engineering, College of Engineering, Nawroz University, Duhok 42001, Iraq; 6CERIS, Civil Engineering, Architecture and Georresources Department, Instituto Superior Técnico, Technical University of Lisbon, Av. Rovisco Pais, 1049-001 Lisbon, Portugal; 7Department of Civil Engineering, College of Engineering, Prince Sattam Bin Abdulaziz University, Alkharj 16273, Saudi Arabia; r.alyousef@psau.edu.sa; 8Structural Engineering Department, Structural Engineering and Construction Management, Future University in Egypt, New Cairo 11835, Egypt; ahmed.deifalla@fue.edu.eg

**Keywords:** supplementary cementitious material, natural fibers, coconut waste, ultrasonic pulse velocity, high strength concrete, shear strength

## Abstract

The issue of brittleness and low post-peak load energy associated with the plain HSC led to the development of fiber-reinforced concrete (FRC) by using discrete fiber filaments in the plain matrix. Due to the high environmental impact of industrial fibers and plasticizers, FRC development is ecologically challenged. Sustainability issues demand the application of eco-friendly development of FRC. This study is aimed at the evaluation of coir as a fiber-reinforcement material in HSC, with the incorporation of silica fume as a partial replacement of cement. For this purpose, a total of 12 concrete mixes were produced by using three different doses of coir (0%, 1%, 1.5%, and 2% by wt. of binder) with silica fume (0%, 5%, and 10% as volumetric replacements of cement). The examined parameters include compressive strength, shear strength, splitting tensile strength, ultrasonic pulse velocity, water absorption, and chloride ion permeability. The scanning electron microscopy (SEM) technique was adopted to observe the microstructure of the CF-reinforced concrete. The results revealed that due to the CF addition, the compressive strength of HSC reduces notably; however, the splitting tensile strength and shear strength experienced notable improvements. At the combined incorporation of 1.5% CF with 5% silica fume, the splitting tensile strength and shear strength of the concrete experienced improvements of 47% and 70%, respectively, compared to that of the control mix. The CF incorporation is detrimental to the imperviousness of concrete. The combined incorporation of CF and silica fume is recommended to minimize the negative effects of CF on the permeability resistance of concrete. The SEM results revealed that CF underwent a minor shrinkage with the age.

## 1. Introduction

In recent years, applications of HSC have risen in the construction sector due to several customized advantages as compared to conventional plain concrete, such as improved workability, higher compressive strength, and excellent durability characteristics. These properties are attained by utilizing appropriate amounts of supplementary cementitious materials (SCMs) and water-reducing admixtures at a low water to binder ratio. The use of waste SCMs has significantly increased in HSC development in recent years as a consequence of increased awareness about environmental issues. Incorporating waste SCMs, e.g., silica fume, ground steel slag, waste glass powder, and fly ash, can substantially reduce the cement content of concrete [1,2,3,4], and subsequently, it produces environmental-friendly concrete. Solid wastes from municipal, industrial, and agricultural fields can be used as SCMs. Reusing solid waste materials or industrial by-products as a partial replacement of cement is a sustainable approach to resolve environmental issues.

Silica fume is a by-product of ferro–silicon alloys owing to its extremely fine particle size, and it contributes to the strength and durability of concrete due to its high pozzolanic potential [5]. The portlandite generated during the hydration of the cement reacts with the fine silica particles and contributes to the strength of concrete, besides filling the space between cement particles to yield high imperviousness [5,6]. Despite environmental and technical benefits, high binder contents with SCMs cause high vulnerability to early-age shrinkage cracking, and this can lead to the service degradation of structural concrete elements [7,8]. The presence of micro-cracks in the matrix of concrete can facilitate the permeability of harmful chemicals (sulfates, chlorides, and carbon dioxide, etc.) inside the concrete structures and it may also increase the brittleness of plain concrete. The brittleness of concrete also increases with an increase in the cement content and the strength class of concrete families [9].

To supplement the tensile strength and toughness of HSC and to control the cracks, fibers have proven to be very encouraging [8,10]. Fiber reinforcement also advances the flexural strength, impact toughness, and energy absorption capacity under both compression and tension by containing the crack proliferation [2,11,12,13]. It also helps in controlling the cracks developed during the shrinkage of high binder contents in HSC families [10,14], thus delaying the onset of failure under different types of loads. Owing to high tensile and flexural strength per unit volume, FRC elements are lighter and more slender as compared to those of plain concrete [15] and they facilitate long-span construction.

Generally, industrial fibers, such as steel fibers, glass fibers, polypropylene fibers, polyvinyl alcohol fibers, basalt fibers, and carbon fibers, have been used for the development of FRC [16,17,18,19,20,21]. The type and dosage of fiber are usually selected based on the application of FRC. Despite offering several ductility benefits, FRC development is expensive due to the requirement of costly fibers and water-reducing admixtures. The environmental impact of FRC at the production end is significantly higher when compared to conventional plain concrete [22]. Ali et al. [23] estimated the FRC containing a 1% volume fraction of steel fiber, glass fiber, and polypropylene fiber, respectively, yielded 60%, 15%, and 5% more carbon footprint compared to that of the plain concrete. Industrial fibers contribute a large portion to the cost and carbon footprint of concrete. In addition to the high cost of fibers, the addition of superplasticizers to control the workability also increases the final cost and footprint of FRC development. This situation under the prevailing environmental challenges demands the application of eco-friendly substitutes of conventional fibers.

There are several inexpensive and eco-friendly alternatives for industrial fibers. These include recycled waste fibers and natural fibers. FRC can be developed with waste fibers derived from the tire and polymer scrapes. The carbon footprint of processing recycled steel fibers from scrap tires is 20 times smaller than that of industrial steel fiber [24]. Recycled steel fiber has demonstrated significant improvement in the ductility behavior of concrete [25,26]. Polymer fibers from scrap tires can also substitute the function polypropylene fiber in HSC [27,28]. Similarly, recycled carbon fibers have exhibited substantial increments in the compressive and flexural behavior of HSC [29]. Meanwhile, researchers have also advanced the research on the engineering performance of concrete with natural fibers. Several natural fibers that have been used by humans for centuries possess very high tensile strength and flexibility, e.g., coir, jute, sisal, Musaceae, and hemp, etc. [30,31,32,33,34]. Multiple options for natural fibers are available in many countries, which can be used for FRC development without seriously affecting the carbon footprint of concrete.

Compared to most natural fibers, coir is generated as a waste product of coconut fruits. The global annual production of coconut is around 60 million tons [35]. Generally, coconut seed is separated from the fruit for edible purposes, and coir and coconut shells are the byproducts. Coir consists of thick bundles of strong natural fibers, also called filaments. It possesses low density, high tensile strength, and high abrasion resistance; thus, it is suitable to make ropes, brushes, mattresses, and mats, etc. [36]. It is one of the few natural fibers that is resistant to saltwater, and can withstand high tensile and flexural deformations, making it suitable for composites subjected to dynamic and elastic deformations [37,38,39]. The tensile strength of coir ranges between 80 MPa and 150 MPa; thus, it can be considered as a substitute for low-density polymer fiber for FRC manufacturing.

Until now, the feasibility of coir as a fiber reinforcement for cementitious materials has been examined by several researchers [40,41,42,43,44]. Ali et al. [45] investigated the dynamic properties of medium strength concrete (MSC) made with coir. It was found that the damping of concrete increased significantly with the increasing coir content. FRC having coir chopped in lengths of 50 mm and at a content of 5% by wt. of cement yielded the best dynamic properties. Baruah and Talukdar [46] studied the properties of normal strength concrete (NSC) with various volume fractions of CF. It was observed that at a 1.5% volume of CF, the compressive strength, splitting tensile strength, flexural strength, and shear strength of plain concrete were improved, respectively, by 17%, 17%, 25%, and 32%. Hwang et al. [40] studied the mechanical properties of different strength grade cementitious composites with coir incorporation. It was found that higher volumes of coir were detrimental to the workability, density, and compressive strength of concrete. However, there was a distinct rise in the flexural capacity and impact resistance of the concrete with coir incorporation.

Khan and Ali [47] reported the effects of adding plasticizer and silica fume on the mechanical properties of coir-reinforced MSC. Synergistic behavior was observed when coir was incorporated with silica fume and plasticizer. It was demonstrated that the compressive and flexural toughness of concrete were substantially improved when coir was used in conjunction with plasticizer and silica fume. Ahmad et al. [31] studied the properties of HSC with varying dimensions (25, 50, and 75 mm) and contents (0.5%, 1%, 1.5%, and 2%) of coir. A maximum increase in the compression and flexural properties was observed with incorporation of 50 mm long coir at 1.5% by wt. of cement. The compressive strength of HSC was observed to be increased by 25% at a 1.5% incorporation of 50 mm long coir. Flexural and splitting tensile strength of the HSC experienced minor improvements due to the coir addition. Sekar and Kandasamy [48] observed the characteristics of NSC with coir and coconut shells. Increasing the incorporation of coir demonstrated moderate increments in the compressive strength and flexural strength of NSC, while a substantial increment in the impact toughness of concrete was observed due to the coir addition with and without coconut shells as aggregate replacements.

Studies relevant to the durability properties are still very few. Hwang [40] studied the water absorption and plastic cracking of coir-reinforced concrete. The water absorption of concrete increased with the rise in coir content and water-binder ratio. However, plastic cracking of cementitious composites was reduced with coir. No plastic cracking was noticed in samples of 2.5% and 4% volume of coir, respectively, at a 0.3 and 0.35 water-binder ratio. Mydin et al. [49] demonstrated a reduction in the porosity and water absorption of lightweight concrete due to the addition of alkali-treated and untreated coir. An increase in ultrasonic pulse velocity also suggested that the density and porosity of concrete were reduced due to the coir addition. Ramli et al. [50] found that the intrinsic permeability and carbonation of HSC increased with the increase in the utilization of the coir reinforcement. This was linked with the low density and shrinkage of coir. Sivaraja et al. [51] reported that the mass loss due to the freeze-thaw in the coir-reinforced concrete was acceptable compared to that of conventional concrete.

An overview of the existing literature demonstrated that fewer investigations [50,52] have been conducted to understand the mechanical and permeability behavior of HSC with coir, as fiber reinforcement has one of the major applications in advancing the ductility of HSC. Moreover, the effect of a mineral admixture used in conjunction with coir should also be investigated on the performance of HSC to maximize sustainability, ductility, and durability benefits. The main objective of this study is to explore the mechanical and permeability characteristics of HSC with silica fume and coir. Therefore, a total of twelve concrete mixes were made by using three different contents of coir (0%, 1%, 1.5%, and 2% by wt. of binder) with and without silica fume (0%, 5%, and 10% as the volumetric replacement of cement). The workability of coir-reinforced mixes was controlled by the manipulation of the superplasticizer dosage. The studied parameters include compressive strength, shear strength, splitting tensile strength, ultrasonic pulse velocity, water absorption, and chloride ion permeability. SEM technology was adopted to observe the microstructure of the CF-reinforced matrix. To the best knowledge of the authors, this is the first study to assess the shear strength and chloride ion penetration of CF-reinforced mixes. The results of this would benefit the development of ecofriendly FRC and promote the application of biowaste fibers in the construction industry.

## 2. Experimental Program

### 2.1. Characteristics of Materials

#### 2.1.1. Binding Materials

Ordinary Portland cement (OPC) was used as the main binder for the preparation of all mixtures. This general-purpose cement complies with ASTM C150 [53], having 53 Grade. This cement is commercially available as a Maple Leaf 53 Grade general-purpose cement suitable for high-performance concretes. Table 1 shows the important properties of OPC. Silica fume, which is a waste of ferrosilicon alloys, in condensed form was obtained from Sika Pvt. Ltd. Lahore, Pakistan. The silica content of silica fume is around 98.5% and it has a specific surface area of 27,000 m^2^/kg. Owing to its extremely small particle size compared to that of the cement, silica fume offers a filling effect and a high pozzolanic potential with OPC. Granulometry of silica fume and OPC is shown in Figure 1.

#### 2.1.2. Aggregates

For fine aggregate, siliceous sand was used. This sand was picked from the Lawrancepur quarry in the Northern Punjab region (situated at 33.8308° N, 72.5057° E) of Pakistan (Attock, Punjab). Owing to well-gradation, this sand is recommended for high-performance concrete development. Dolomitic sandstone was used as coarse aggregates. This high-quality aggregate was obtained from the Kirana Hills quarry situated in the central Punjab region of Pakistan (Sargodha, Punjab, 31.9667° N, 72.7029° E) was used as coarse aggregate. For the development of high-performance concrete, the maximum aggregate size of the coarse aggregate was kept smaller than 12.5 mm. The characteristics of both aggregates are provided in Table 2. Gradation of both types of aggregate is shown in Figure 2.

#### 2.1.3. Water and Superplasticizer

Tap water of concrete laboratory, free from organic matters, was used for the mixing and curing of all batches. The water has a pH of 7.9. Whereas, to control the workability of CF-reinforced mixtures, Sika Viscocrete 3110 was used. This belongs to the polycarboxylate-based 3rd generation of water-reducers. It complies with the ASTM C494 [55] as a type G plasticizer.

#### 2.1.4. Coir or Coconut Fiber

Coir was extracted from waste coconut husks and chopped manually into lengths of 50 mm. This length of coir was selected based on its optimum effect on the mechanical performance reported in the literature [31,45]. The color of the coir was light tan, and its diameter was around 0.1 mm as assessed from the SEM observation, as shown in Figure 3. According to several studies [56,57,58], the axial tensile strength of coir ranged between 47–178 MPa. The tensile strength of coir is also suspected to vary among the filaments of the same parent fruit due to natural imperfections.

### 2.2. Details about the Preparation of Concrete Mixes

A total of twelve concrete mixes were designed in this study to examine the singular and combined effects of silica fume and coir on the properties of HSC, as shown in Table 3. The first mix served as the control mix; its designated compressive strength, over 50 MPa at the age of 28 days, was selected after performing various trials under the guidance of literature [2,59]. Even though there is no specific boundary between normal strength concrete and HSC, the American Concrete Institute (ACI) conservatively defined HSC as concrete with 28-days’ cylindrical compressive strength above 42 MPa [60]. Moreover, in Pakistani studies [31,61,62], HSC is referred to as concrete having compressive strength from 45 MPa to 60 MPa. The workability of high-performance HSC is usually higher than conventional mixes; therefore, the design of the control mix was chosen for the slump of 200 ± 20 mm [63]. In the control batch, silica fume was incorporated as 0% (SF0), 5% (SF5), and 10% (SF10) by volume replacement of OPC. With each level of silica fume, coir was incorporated at four different percentages, i.e., 0% (CF0), 1% (CF1), 1.5% (CF1.5), and 2% (CF2). For varying percentages of SF, the dosage of CF was taken as wt. of binder in the SF0 family. Using this method, the constant quantity of CF was attained for each percentage of SF (0%, 5%, and 10%). These contents of coir were selected based on their optimum effects on the mechanical performance of concrete [31,56]. Coir contents higher than 2% demonstrated negative effects on workability and strength. With the incorporation of silica fume and coir, the workability of concrete was reduced; thus, the superplasticizer dosage was modified to control the workability of mixes. All mixes yielded slump values in the range of 150–210 mm. Mixes containing 1% and 1.5% coir qualified for high-performance concrete based on their high workability, as no segregation was observed and slump values were well between 180–220 mm [63]. Meanwhile, mixes containing 2% coir exhibited slump values relatively lower compared to the plain mix family. The tangling effect became dominant at 2% coir content; thus, the slump value could not be increased beyond 156 mm, despite increasing the plasticizer dosage. To avoid bleeding, the dosage of the plasticizer was not increased further to achieve workability in the case of the 2% coir content.

All mixtures were prepared in a rapid-speed mechanical mixer equipped with a speed regulator. Firstly, the binding materials and aggregates were mixed together for 2 min at a rotary speed of 30 rpm. Then, the required amount of water and plasticizer was added to the dry mixture, and mixing was done at the speed of 60 rpm for 3 min. In the case of plain concrete mixtures, the freshly mixed concrete was tested for a slump and proceeded for casting. Whereas, in the case of fiber-reinforced mixes, the coir was gradually added to the fresh concrete in the running mixer to ensure uniform dispersion of fibers. The mixing speed during the addition of coir was increased to 80 rpm and this stage lasted for about 2 min. The high-speed mixing ensured a thorough mixing of coir filaments without accumulation. After completion of mixing, the samples were tested for slump testing and subsequently for casting. The schematic representation of the mixing procedure is illustrated in Figure 4.

### 2.3. Experimental Testing Methods

A CONTROLS Universal Testing Machine of 3000 kN capacity was used to perform the compressive strength test on 100 mm × 100 mm × 100 mm cubical samples according to BS: EN 12390 [64]. The compressive strength of each mix was determined at the age of 28 and 90 days. To understand the ductility of coir-reinforced mixes, shear strength and splitting tensile strength tests were conducted. A bi-surface shear strength (*B**_s_*) test was conducted on the specimens of 150 mm × 150 mm × 150 mm dimensions, as shown in the schematic setup in Figure 5. This testing technique was adopted from the literature [65]. *B**_s_* is calculated using Equation (1). Where *P_u_* is the peak load (kN), *A**_s_* = area of shear plane. For the determination of the splitting tensile strength, 100 mm × 200 mm cylindrical samples were tested in a CONTROLS Universal Testing Machine as per ASTM C496 [66]. The shear strength and splitting tensile strength of each mix was determined at the age of 28 days.
(1)$$B_{s}=\frac{P_{u}}{2A_{s}}$$

Ultrasonic pulse velocity test was performed on 100 mm cubical samples of all mixes following ASTM C597 [67] to assess the effect of silica fume and coir on the homogeneity and density of the concrete mix. For the assessment of permeable pore volume, the water absorption capacity of all concrete mixes was determined. For this purpose, 100 mm diameter × 50 mm thick samples were tested to determine the absorption capacity according to ASTM C948 [68]. Chloride permeability in all mixes was estimated by using the immersion method [69,70]. Concrete samples cured for 28 days were soaked in a 10% sodium chloride (NaCl) solution for a period of 90 days. After conditioning in chloride solution for 90 days, the samples were split into halves using a compression load. The failed surfaces of the chloride-conditioned samples were sprayed with a 0.1 normality solution of silver nitrate (AgNO_3_). When the nitrate solution reacted with the penetrated chlorides, it left silver-colored precipitates forming silver chloride (AgCl), also indicating the penetration depth of chloride ions, as shown in Figure 6. The depth of chloride penetration was recorded at six different points and the average value was taken as the representative value of the sample. In this study, three replicate samples of each mix were tested to determine a parameter. The results with the standard deviation values between the three samples are presented in the results and discussion section.

## 3. Results and Discussion

### 3.1. Compressive Strength

The effect of the coir addition with and without silica fume on the compressive strength of HSC is shown in Figure 7. The net change in the compressive strength with the variation in the silica fume and coir content is depicted in Figure 8. The compressive strength of HSC improved significantly due to the partial replacement of cement with silica fume. As the silica fume possesses pozzolanic potential, it can help in consuming free portlandite in the binder matrix and transforming it into a dense calcium silicate hydrate gel [71,72]. The filler effect of mineral admixtures improves the density of microstructure [73]; ultimately, the strength of the HSC is improved. The compressive strength was improved by around 10% and 16%, respectively, at 5% and 10% replacement of cement with silica fume. At the age of 90 days, concrete mixes containing 10% silica fume demonstrated the highest compressive strength values for a given coir content. The maximum strength gain between 28 and 90 days was observed for mixes containing 10% silica fume. This is because the pozzolanic reaction between the micro-silica particles and portlandite is slow; therefore, concrete mixes made with the silica fume develop strength over a long period and demonstrate high net gains in the strength between 28 and 90 days.

The incorporation of coir fibers led to notable reductions in the compressive strength of HSC. Compressive strength was reduced by about 4%, 15%, and 17%, respectively, due to the incorporation of 1%, 1.5%, and 2% coir. These reductions can be blamed on the two factors: (1) the density of coir is considerably smaller compared to the concrete and binder matrix; (2) fiber accumulation may increase the porosity of concrete. Fiber-reinforcement is not as effective in compression as it is in splitting tensile strength and flexural strength. Similar to coir, industrial polypropylene and polyvinyl fibers, due to their lower density, have also exhibited negative effects on the compressive strength [15,74,75].

Low-density fibers may also act as the voids in concrete, such that increasing the coir content is anticipated to increase the porosity of concrete. Previous studies have confirmed both positive and negative effects of coir on the compressive strength [31,76]. The findings of this study are in agreement with Ramli et al. [50], who studied the HSC with different contents of coir. Meanwhile, the positive effects of coir were observed on the compressive strength of lightweight concrete [77,78]. As lightweight concrete (with densities of 1050 and 1350 kg/m^3^) is not sensitive to the incorporation of low-density coir content, thus the confinement effect of fibers demonstrated an increase in compressive strength. It is also postulated that the HSC family is highly sensitive to the strength reduction owing to the coir addition as compared to lower strength class families, i.e., MSC, NSC, lightweight concrete, etc. Silica fume controlled some of the negative effects of coir addition on the compressive strength of HSC. This could be credited to the strengthening of the concrete matrix owing to the pozzolanic reactions. Silica fume may also solidify the bond between the fibers and concrete matrix.

### 3.2. Shear Strength

The effect of varying contents of coir and silica fume on the bi-surface shear strength of HSC is illustrated in Figure 9. The results demonstrated that both silica fume and coir positively affected the shear strength of concrete. The incorporation of 5% and 10% silica fume led to a 14.9% and 8.7% increment in the shear strength of plain concrete. These improvements have been credited to the microstructural growth caused by the reaction between portlandite and high active micro-silica particles. The silica fume addition influences compressive and tensile properties similarly, as both properties mainly depend on the development and density of the microstructure.

The incorporation of coir proved to be extremely useful in enhancing the shear strength of HSC. The shear strength of SF0/CF0 or control mix was increased by 39.3%, 59%, and 45.5%, respectively, at 1%, 1.5%, and 2% contents of coir. As coir filaments possess high tensile strength, these can effectively supplement the shear resistance of the plain HSC matrix. Similar to industrial fibers, i.e., glass, polypropylene, and steel fibers [79,80], coir is also beneficial to the shear strength of the plain matrix. However, the contribution of coir to the shear strength is smaller compared to those observed in the case of industrial fibers [79]. This could be because of the weaker bond between organic fibers and binder compared to the bond of industrial fiber with the binder. Fiber reinforcement did not only improve the peak strength of concrete but also increased the energy absorption capacity under the shearing stress [80].

The shear strength of HSC was improved by more than 70% (w.r.t SF0/CF0), due to the incorporation of 5% silica fume and 1.5% coir. The results of the mixes made with the combined incorporation of silica fume and coir demonstrated that the benefits of coir and silica fume can be combined to obtain a higher shear strength. However, synergistic behavior was not observed due to the combined incorporation of coir and silica fume.

The relationship between the compressive strength (f_c_) and bi-surface shear strength (*B**_s_*) is shown in Figure 10. The correlation of *B**_s_* with f_c_^0.5^ was derived as the function of coir percentage (CP). As the ratio between f_c_ and *B**_s_* (i.e., *B**_s_*/f_c_) for plain and coir-reinforced mixes are significantly different, the relationship between these two parameters was derived as the function of CP. The value of *B**_s_*/f_c_^0.5^ also indicates the ductility of concrete. The increase in the *B**_s_*/f_c_^0.5^ value with the rising CP implies an increase in ductility. A strong linear correlation is found between CP and *B**_s_*/f_c_^0.5^, as the coefficient of the determination value is 0.827. Thus, using f_c_, the value of *B**_s_* can be predicted with good accuracy at a required CP.

### 3.3. Splitting Tensile Strength

The effect of coir and silica fume incorporation on the 28-day splitting tensile strength of HSC is shown in Figure 11. The incorporation of silica fume exhibited minor improvements of 5–9% in the splitting tensile strength. Plain mixes containing 5% silica fume demonstrated higher tensile strength than those containing 10% silica fume. A higher percentage of silica fume demonstrates higher strength at the later ages, as shown in the results of compression testing at 90 days. Generally, the incorporation of silica fume is useful to the tensile strength of concrete owing to the pozzolanic reactions between free portlandite and micro-silica particles.

With the variation in dosage, the coir reinforcement exhibited alternating effects on the splitting tensile strength. The incorporation of 1% and 1.5% coir contents improved the tensile strength of plain HSC by 6% and 4%, respectively. Whereas, the 2% coir exhibited the reduction in the tensile strength value of w.r.t plain HSC. The positive effect of low contents of coir on the tensile strength can be credited to a nominal increase in the crack-bridging capacity of concrete. Whereas, at higher contents, the splitting tensile strength of HSC decreased, probably due to the increasing issue of the accumulation of coir filaments. This can significantly reduce the porosity of concrete, as well as the efficiency of fiber reinforcement. Ahmad et al. [31] observed an optimistic net gain of 20% in the splitting tensile strength of HSC at 1.5% coir content. Literature [50,81] also confirmed the negative effects of higher contents of coir on the short-term and long-term mechanical properties of concrete.

It is worth mentioning here, that similarly to synthetic fibers, the coir reinforcement shows a positive influence on the ductility of concrete. Unlike plain HSC, the failure of coir-reinforced mixes is delayed by the bridging effect of fibers, as shown in Figure 12. Despite nominal contributions towards the peak tensile strength, coir advances the ductility of concrete by delaying the onset of failure. The residual tensile strength of fiber-reinforced HSC was notable after the application of the peak load.

The mixes made with the combined addition of silica fume and coir demonstrated superior tensile strength compared to the mixes with the individual incorporation of silica fume and coir. Furthermore, the net effect of combined incorporation of silica fume and coir on splitting tensile strength was greater than the sum of net effects due to the individual incorporation of silica fume and coir, as shown in Figure 11b. The bond strength of coir with concrete is anticipated to be improved due to the strengthening of the binder matrix with the silica fume addition. The dispersion of fibers may also be improved with the use of silica fume as a partial replacement of cement. The efficiency of coir reinforcement increased with the increasing level of silica fume. The fiber-reinforced mixes containing 10% silica fume yielded higher tensile strength than the mixes with 5% silica fume. The increase in the dispersion of fibers and bond strength with the silica fume addition improves the transfer of tensile stresses from the plain concrete matrix to the fiber reinforcement [82].

### 3.4. Ultrasonic Pulse Velocity (UPV)

Indirect assessment of the quality of concrete is usually done by conducting a UPV test. The values of UPV can be linked with the accurate strength and density of concrete. UPV values ranging between 3.5 km/s and 4.5 km/s indicate a good quality concrete, while UPV values above 4.5 km/s indicate excellent quality of concrete [83]. Usually, HSC concrete families exhibit UPV values for excellent quality. The results of UPV testing are illustrated in Figure 13. Silica fume improves the quality of concrete by advancing its strength and density. Thus, all mixes incorporating silica fume with and without coir exhibited UPV values higher than 4.5 km/s. Fine particles of silica fume increase the CSH gel responsible for the strength enhancement and reduce the porosity of concrete by the filler action. This leads to a faster propagation of ultrasonic waves through the concrete media.

A decline in the UPV value of HSC was noticed with the increasing coir content. These reductions in the UPV can be primarily caused by the introduction of a low-density material in concrete. Furthermore, the increasing number of interfacial transition zones (ITZs) within a concrete mass is also found to negatively influence the UPV value [84]. The UPV value reduces by 10% when 2% coir content is used in HSC. This demonstrates that the incorporation of coir can reduce the quality of concrete.

The incorporation of low-density synthetic fibers has also demonstrated a minor decline in the UPV of concrete [85]. Das et al. [86] showed that the incorporation of a 1% volume fraction of polypropylene fiber reduces the UPV of NSC by 3.2%. However, in this study, a higher degradation in the UPV was observed because of the absorbent nature of coir. During the mixing, coir can absorb water from the fresh concrete, and this may cause shrinkage problems within the filament of fibers. The reductions in the size of filaments can increase the porosity and micro-channels inside the concrete. Thus, the coir incorporation demonstrated a negative effect on the UPV of HSC. The silica fume minimized or compensated the loss in UPV due to coir incorporation. Owing to the silica fume addition, mixes made with 1.5% and 2% coir exhibited UPV values corresponding to excellent quality. The positive effect of silica fume on the UPV and quality of cementitious materials is also reported in the literature [87,88]. As both compressive strength and UPV are related to the density and the hydration of cementitious compounds, both of these parameters can be correlated with high accuracy, as shown in Figure 14. The change in the density and porosity of concrete due to the incorporation of silica fume and coir similarly affects the compressive strength and UPV; thus, both of these parameters are linearly proportional to each other with a good value of the coefficient of determination (R^2^ > 0.85).

### 3.5. Water Absorption Capacity

The durability of concrete structures is directly affected by the transport of water into the microstructure of concrete. The water absorption capacity indicates the permeable porosity of concrete. The permeable pore volume in concrete is affected by the connectivity and tortuosity of pores inside the concrete media. A water absorption test was conducted on the samples cured for 28 days. The effect of the different contents of silica fume and coir on the water absorption capacity of concrete is shown in Figure 15.

It was observed that the increase in silica fume content from 0% to 10% reduced the water absorption capacity of HSC by 32%. The decrease in the water absorption is due to the decline in the connectivity of pores. Silica fume refines the size of pores and increases the tortuosity between pore connections. This ultimately reduces and slows the absorption of water into the microstructure of concrete. Furthermore, the reaction between free portlandite and silica generates dense CSH gel to enhance the impenetrability of concrete. The water absorption capacity of concrete was increased by 5%, 11%, and 33% due to the incorporation of 1%, 1.5%, and 2% coir by wt. of binder, respectively. Fiber reinforcement is known to increase the permeability and absorption of concrete due to the increased interaction between pores [89,90]. Although fiber reinforcement provides control over the drying, shrinkage and cracking in the binder matrix [10], it can also increase the porosity due to the accumulation of filaments inside the concrete.

Owing to the positive effect of silica fume, the water absorption of coir-reinforced mixes is significantly controlled. All mixes containing silica fume with or without coir fibers exhibited water absorption capacity more than that of the plain HSC. The growth of more CSH gel due to a pozzolanic reaction can reduce the gap between the fiber filaments and matrix, consequently leading to the restriction of water transport.

### 3.6. Depth of Chloride Ion Penetration

The permeability of chloride ions in concrete causes the corrosion of steel rebars in reinforced concrete structures. Thus, it is convenient to study the chloride ion permeability of concrete families for the assessment of durability. The permeability of chloride ions into concrete is highly influenced by similar factors that influence water permeability, i.e., pore size, pore connectivity, etc. The effect of varying coir and silica fume contents on the chloride ion penetration depth (CPD) of HSC concrete is shown in Figure 16.

It was noticed that the incorporation of coir increased the CPD value. An increase of 66% in the CPD was observed when 2% coir reinforcement was incorporated into the plain HSC. As coir reinforcement makes the HSC more permeable, it allows a deeper penetration of the chloride-bearing solution into the concrete. Unlike coir reinforcement, silica fume incorporation drastically reduced the CPD. The incorporation of 5% and 10% silica fume by mass of cement reduced the CPD by 50% and 63%, respectively. The CPD value reduces, as the porosity and permeability of concrete are reduced by the silica fume incorporation. The literature [1,89,91] has exhibited that the incorporation of mineral admixtures is highly useful in enhancing the permeability-related durability properties of concrete. Besides improving the long-term mechanical strength, the mineral admixtures owing to the filler effect demonstrate a positive effect on improving the imperviousness of concrete from an early age. Thus, silica fume also controls the negative effects of coir on the CPD of HSC.

### 3.7. SEM Image Analysis

SEM images of coir reinforcement in the binder paste were analyzed, see Figure 17. The small chunks of hardened binder pastes containing coir were derived from the samples cured for 56 days in tap water. It can be observed that the coir filaments exhibited little or negligible change in volume upon drying. Under the crushing action on the binder sample, some fibers exhibited breaking failure and some exhibited a bonding failure. As can be noticed, some coir filament remained undamaged after extraction; this was the indication that the binder matrix was weaker than the coir, and the coir filament had a higher tensile strength or a short embedded length. Meanwhile, coir can also be observed splitting due to the action of tensile loading on the paste samples. The variance in the failure forms of coir filaments can also be caused by the variation in the embedment lengths of fibers across a contained crack. As under the most common circumstances, the embedment length of fiber is divided unevenly across a crack, thus the failed surface with a long embedment length will tend to cause a splitting fracture of fiber. Whereas the fractured surface containing a short length of filament will most possibly exhibit a pullout failure. The observation of ITZ between coir filament and hardened paste demonstrated that there was little space between fiber and matrix. This little gap is suspected to have been developed due to the pulling action of fibers or due to the minor shrinkage of fibers. SEM images also showed that the crack propagation is stopped by the introduction of coir. This proves that coir reinforcement is important to the fracture toughness of HSC.

## 4. Conclusions

In this study, the effect of varying contents of coir and silica fume was investigated on the properties of HSC. The following are the main conclusions of this study:The coir reinforcement demonstrated a negative effect on the compressive strength of HSC. Silica fume helped in controlling the negative effect of coir on the compressive strength of concrete.The shear strength of HSC improved drastically with the addition of both silica fume and coir. The maximum increase of about 45.5% in shear strength of HSC was observed upon the addition of 2% coir without silica fume. With 5% silica fume and 1.5% coir, the shear strength of HSC was increased by 70% compared to that of the plain HSC.The splitting tensile strength demonstrated a nominal change with the varying contents of coir. A maximum splitting tensile strength, about 6% higher than that of the plain HSC, was observed at 1% coir content. Higher contents of coir negatively affected the splitting tensile strength.The addition of silica fume with coir exhibited synergistic behavior in the results of splitting tensile strength, as silica fume improves the bond strength of fiber filaments by strengthening the binder matrix.The incorporation of low-density coir in the plain concrete reduces the UPV. However, CF-reinforced mixes containing silica fume exhibited higher UPV values compared to those without silica fume.The incorporation of coir drastically increased the water absorption and CPD in plain HSC. Silica fume controlled the negative effects of coir incorporation on the permeability-resistance of HSC.SEM observations indicate a minor shrinkage in coir filaments, which might have increased pore connectivity. Furthermore, both pullout and breaking types of failures were observed for coir filaments.Based on the results of mechanical performance, 1% coir by wt. of binder can produce higher net gains in tensile and shear strength. For further enhancement of mechanical and durability performance of coir-reinforced HSC, the inclusion of 5% silica fume yields the best results.

## Figures and Tables

**Figure 1 materials-15-02616-f001:**
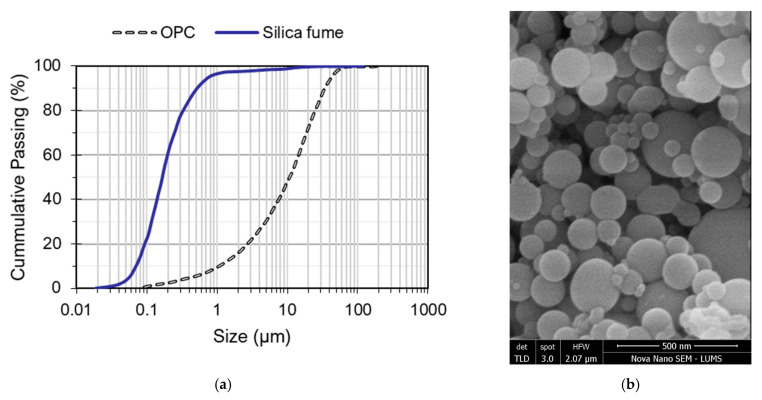
Particle size distribution: (**a**) granulometry of binder materials; (**b**) SEM observation of silica fume particles.

**Figure 2 materials-15-02616-f002:**
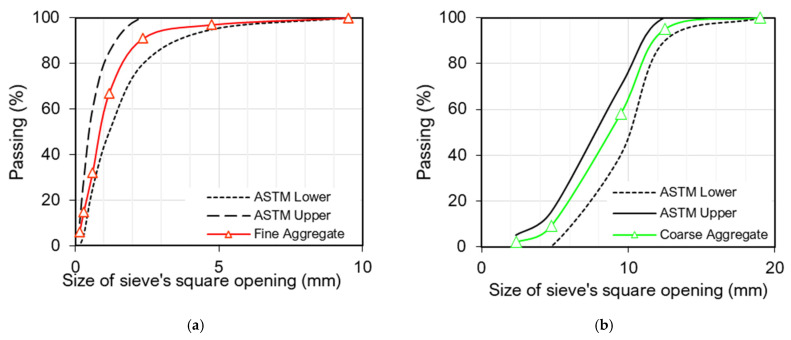
Aggregate size gradation charts within limits of ASTM C33 [54]: (**a**)Fine aggregate; (**b**) Coarse aggregate.

**Figure 3 materials-15-02616-f003:**
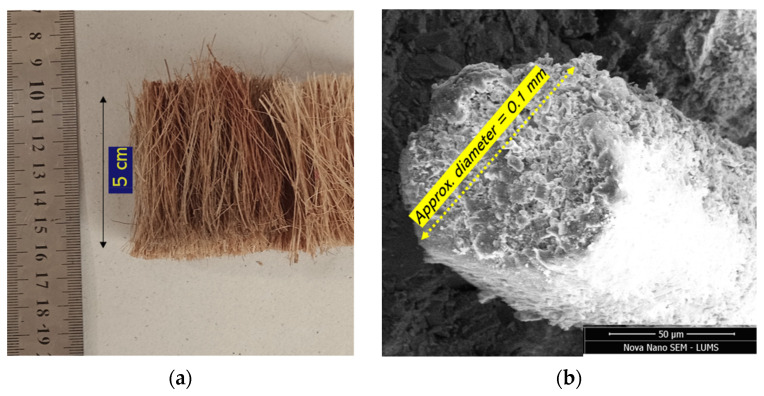
Examination of coir dimensions: (**a**) naked-eye observation; (**b**) SEM observation.

**Figure 4 materials-15-02616-f004:**
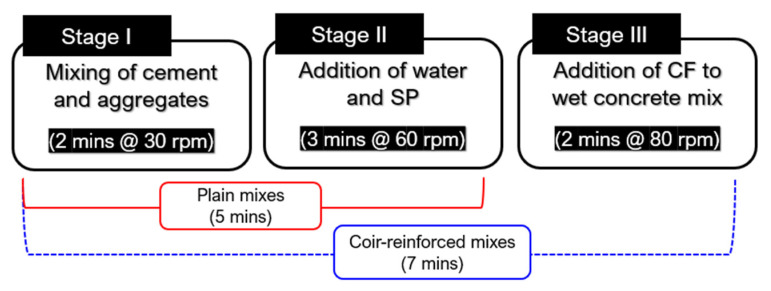
Mixing sequence for the preparation of concrete mixes.

**Figure 5 materials-15-02616-f005:**
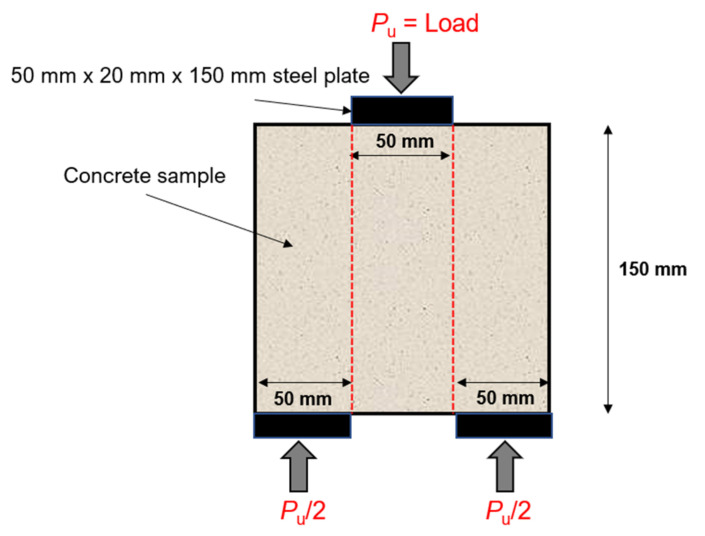
Schematic representation of bi-surface shear test.

**Figure 6 materials-15-02616-f006:**
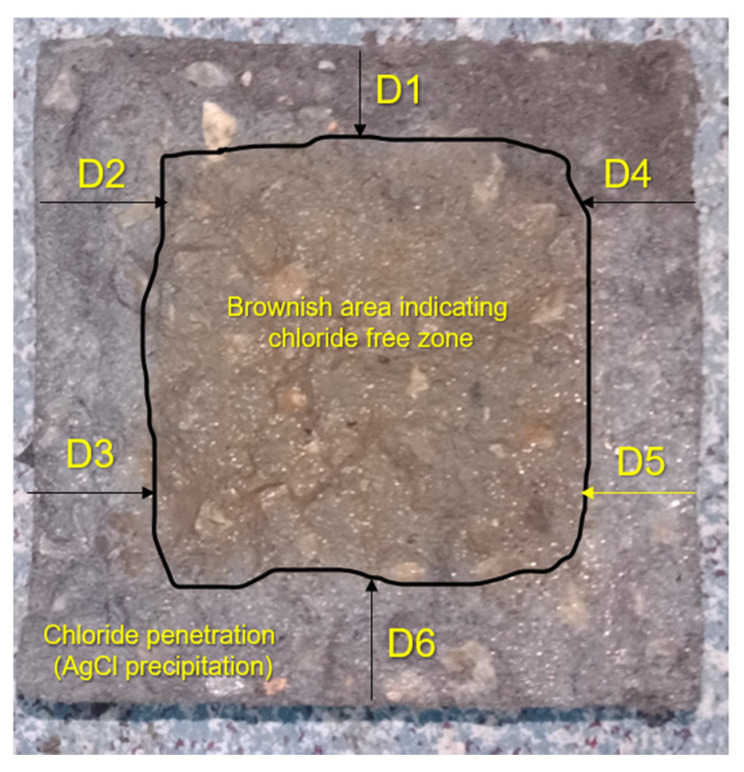
Chloride ion penetration in the concrete sample.

**Figure 7 materials-15-02616-f007:**
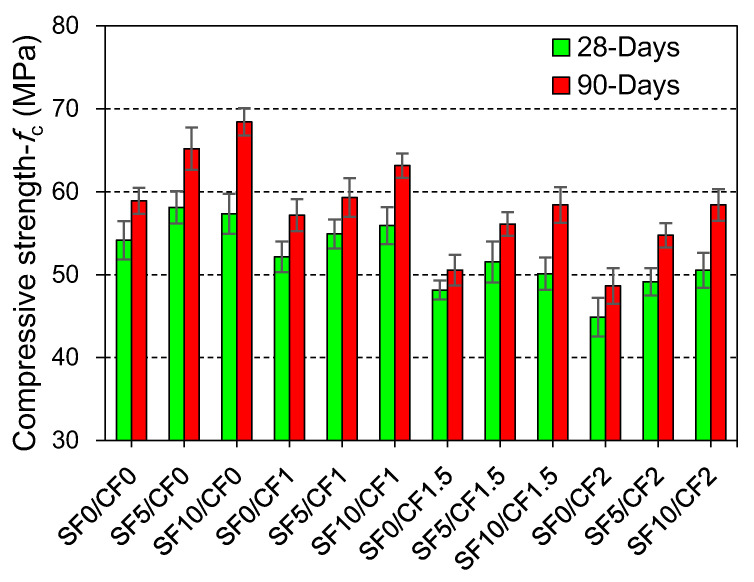
Effect of silica fume and coir addition on the compressive strength of HSC at 28 and 90 days.

**Figure 8 materials-15-02616-f008:**
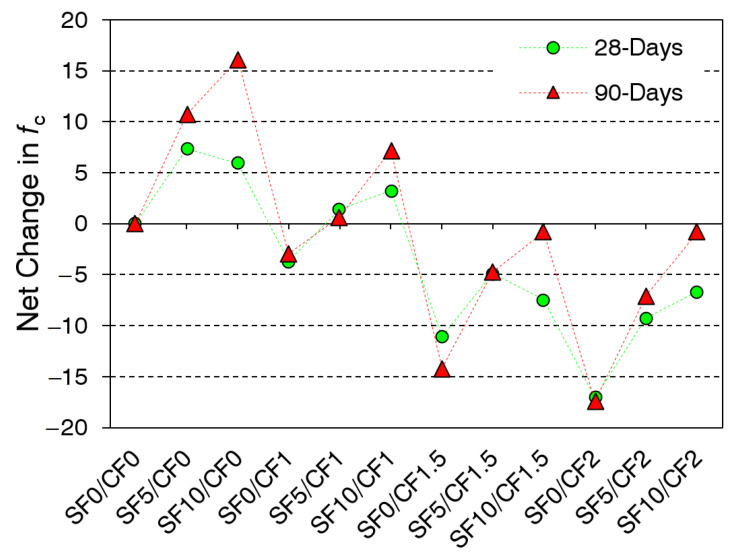
Net change in compressive strength due to silica fume and coir addition.

**Figure 9 materials-15-02616-f009:**
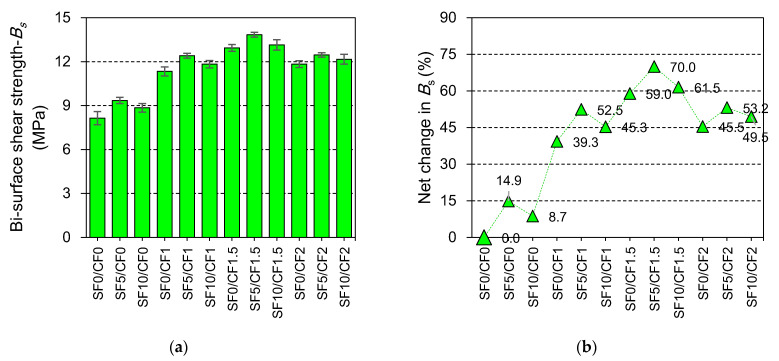
Effect of silica fume and coir incorporation on (**a**) 28 days’ bi-surface shear strength of HSC; (**b**) net change in the bi-surface shear strength of HSC compared to SF0/CF0.

**Figure 10 materials-15-02616-f010:**
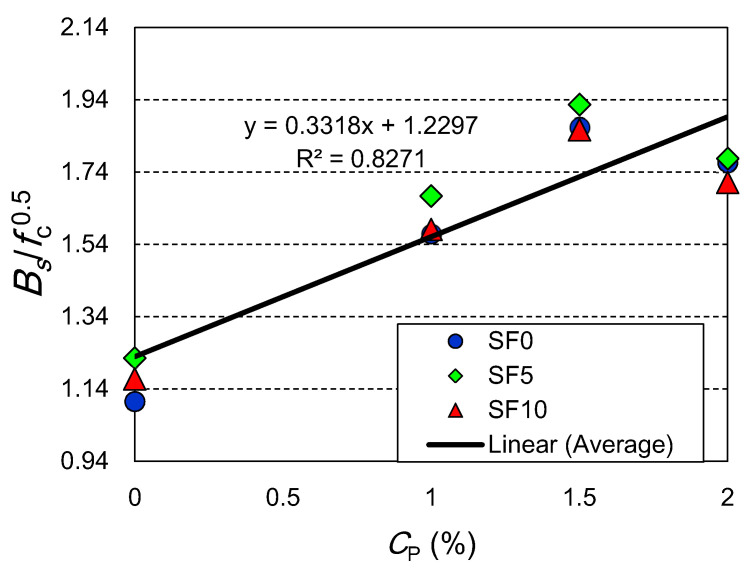
Relationship between shear strength and compressive strength as a function of *C*_P_ (%).

**Figure 11 materials-15-02616-f011:**
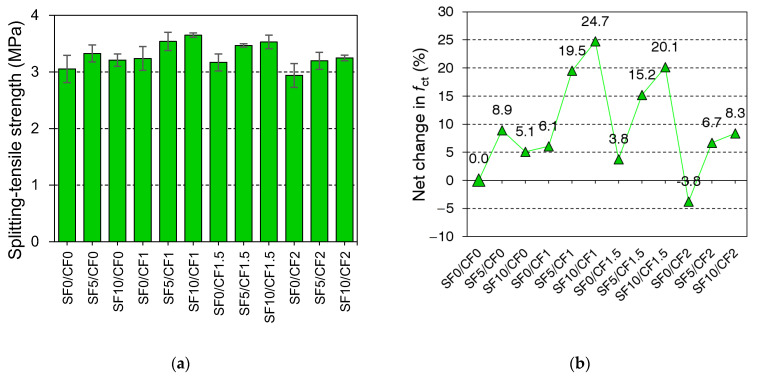
Effect of silica fume and coir incorporation on (**a**) 28-day splitting tensile strength of HSC; (**b**) net change in the splitting tensile strength (*f*_ct_) of HSC compared to SF0/CF0.

**Figure 12 materials-15-02616-f012:**
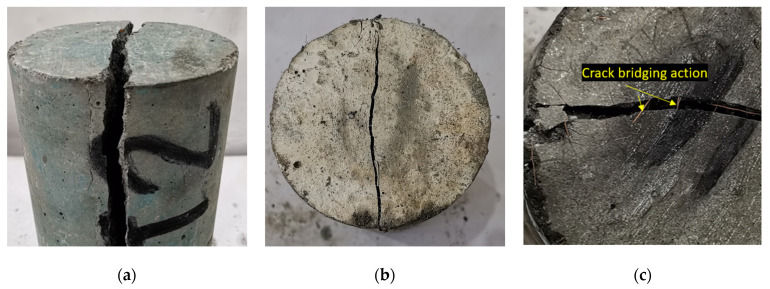
Overview of splitting tensile failure: (**a**) plain HSC; (**b**) HSC with 1.5% coir; (**c**) crack-bridging action of coir.

**Figure 13 materials-15-02616-f013:**
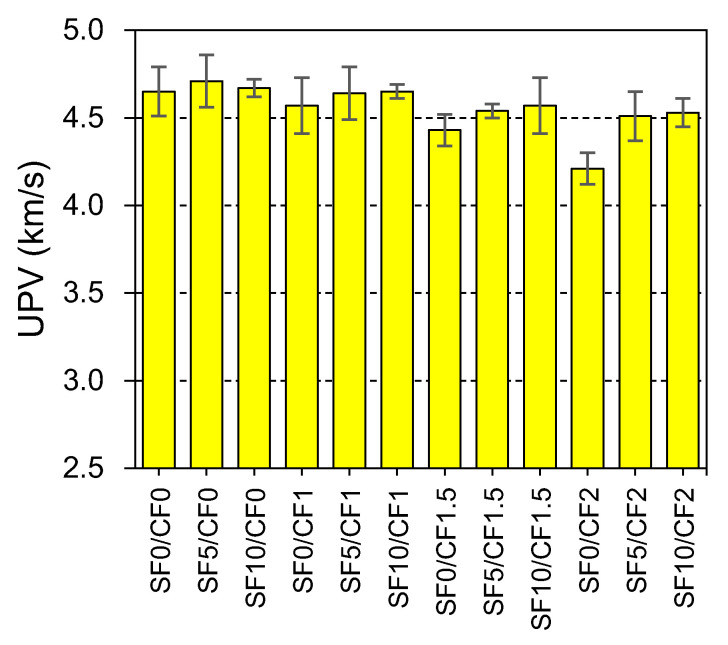
Effect of silica fume and coir reinforcement on the ultrasonic pulse velocity (UPV) of HSC.

**Figure 14 materials-15-02616-f014:**
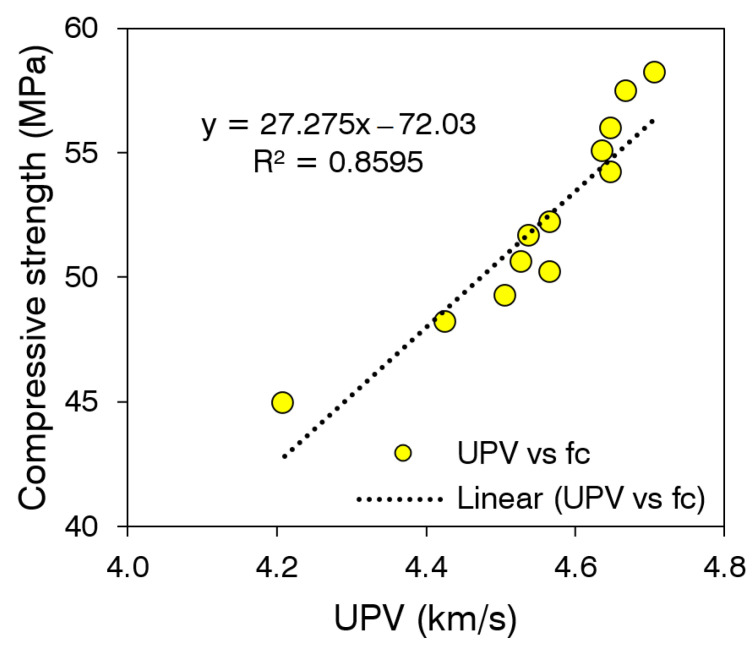
Correlation between corresponding values of UPV and compressive strength.

**Figure 15 materials-15-02616-f015:**
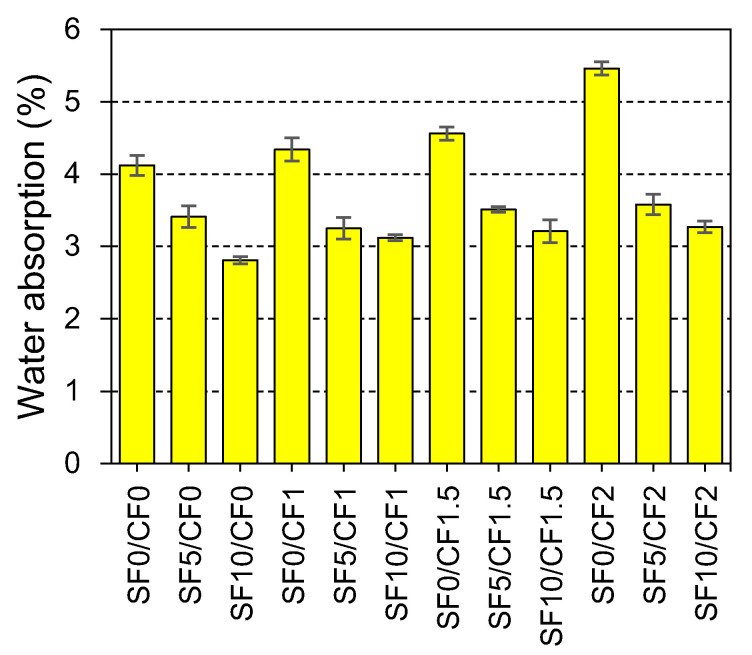
Effect of silica fume and coir reinforcement on the water absorption capacity of HSC.

**Figure 16 materials-15-02616-f016:**
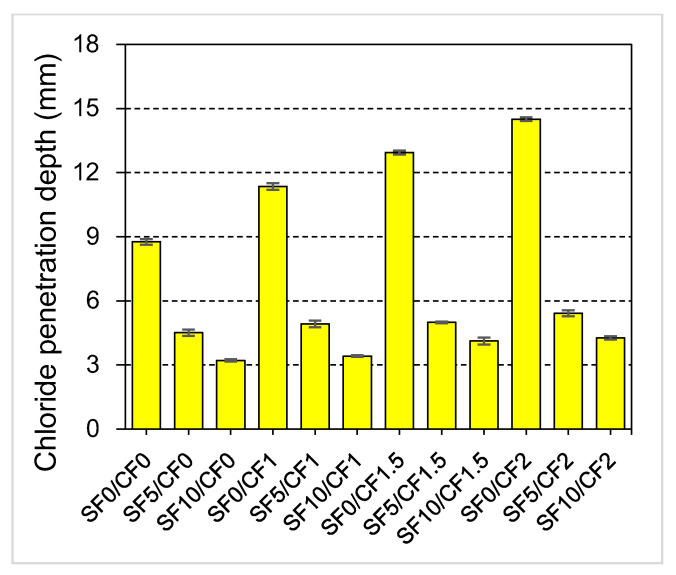
Effect of silica fume and coir reinforcement on the chloride ion penetration of HSC.

**Figure 17 materials-15-02616-f017:**
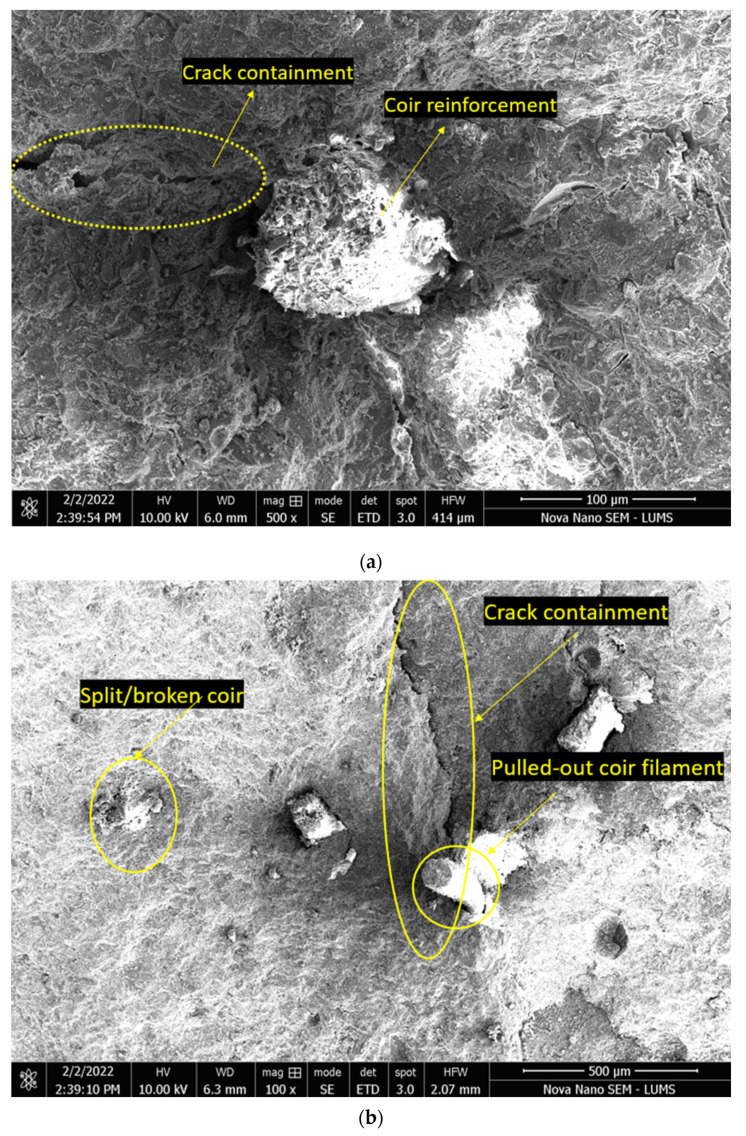
SEM image observations of coir embedded in the binder paste: (**a**) 100 µm; (**b**) 500 µm.

**Table 1 materials-15-02616-t001:** Properties of OPC.

Properties	Name	Result
Chemical composition	Lime-CaO (%)	64.6
	Silica-SiO_2_ (%)	20.4
	Alumina-Al_2_O_3_ (%)	7.3
	Magensia-MgO (%)	3.2
	Loss on Ignition-LOI (800 °C)	1.4
Physical characteristics	Specific-surface area (m^2^/kg)	368
	Specific-gravity	3.12
	Bulk-density (kg/m^3^)	1443
Mechanical strength	Compressive strength at 7 days (MPa) (water-cement ratio 0.29)	48

**Table 2 materials-15-02616-t002:** Properties of aggregates.

Property Name	Fine	Coarse
Maximum particle size (mm)	4.75	12.5
Minimum particle size (mm)	0.075	4.75
Water absorption (%)	0.85	0.96
Specific gravity	2.68	2.71
Dry-rodded density (kg/m^3^)	1685	1655
Fineness modulus	2.91	-
Material	Siliceous-sand	Dolomitic-sandstone

**Table 3 materials-15-02616-t003:** Nomenclature and design of mixtures.

Serial. No.	Mix IDs	CF (%)	SF (%)	OPC (kg/m^3^)	SF (kg/m^3^)	Aggregates		Water (kg/m^3^)	CF (kg/m^3^)	SP (kg/m^3^)	Slump (mm)
						Fa (kg/m^3^)	Ca (kg/m^3^)				
1	SF0/CF0	0	0	475.0	0.0	650	1079	166.3	0.0	2.38	205
2	SF5/CF0	0	5	451.3	18.4	650	1079	166.3	0.0	2.47	209
3	SF10/CF0	0	10	427.5	36.8	650	1079	166.3	0.0	3.13	204
4	SF0/CF1	1	0	475.0	0.0	650	1079	166.3	4.8	2.61	189
5	SF5/CF1	1	5	451.3	18.4	650	1079	166.3	4.8	2.71	194
6	SF10/CF1	1	10	427.5	36.8	650	1079	166.3	4.8	3.44	196
7	SF0/CF1.5	1.5	0	475.0	0.0	650	1079	166.3	7.1	2.85	185
8	SF5/CF1.5	1.5	5	451.3	18.4	650	1079	166.3	7.1	2.96	181
9	SF10/CF1.5	1.5	10	427.5	36.8	650	1079	166.3	7.1	3.75	184
10	SF0/CF2	2	0	475.0	0.0	650	1079	166.3	9.5	2.97	154
11	SF5/CF2	2	5	451.3	18.4	650	1079	166.3	9.5	3.08	156
12	SF10/CF2	2	10	427.5	36.8	650	1079	166.3	9.5	3.91	151

**Note:** CF = coir; SF = silica fume; OPC = ordinary Portland cement; Fa = fine aggregate; Ca = coarse aggregate; SP = superplasticizer.

## Data Availability

Not applicable.

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
