# Peer review of "Development of Ductile and Durable High Strength Concrete (HSC) through Interactive Incorporation of Coir Waste and Silica Fume"

_materials, 2022, doi:10.3390/ma15072616_

Round 1

Reviewer 1 Report

1. In the Introduction, PVA fibers [1] and polyethylene fibers [2] are ignored, which are also widely used in cementitious composites.
Ref.:
[1]Effects of High Temperature on the Burst Process of Carbon Fiber/PVA Fiber High-Strength Concretes. MATERIALS,2019,12(6): 973.
[2]Molecular dynamics study of an amorphous polyethylene/silica interface with shear tests. Materials, 2018, 11(6): 929.
2. If possible please give more information about the used coir or coconut fiber in the research.
3. L206 If possible, please provide the axial tensile strength of the coir in the research since the range is wide. 
4. Why do you choose silica fume in the research? If you want to consider the filling effect of silica fume, you may also use other filling materials, like wollastonite [1], limestone powder [2]. Please add the references and explain.
 Ref.:
 [1]Effects of cations in sulfate on the thaumasite form of sulfate attack of cementitious materials. CONSTRUCTION AND BUILDING  MATERIALS, 2019, 229, 116865. 
5. Table 3 What is the meaning of "Sr."? 
6. Table 3 No. 10, No. 11 and No. 12 have the same 9.5 kg/m3 CF (i. e., 2%) with different OPC. Why? How to calculate the percentage in the research?

Author Response

Dear reviewer,
Enclosed please see the revision of the manuscript and our responses for your valuable comments.

Reviewer 2 Report

The manuscript under review concerns current research and, in a sense, civilization problems (in terms of the possibility of reducing the carbon footprint of concrete composites through the use of additives in the form of natural fibers). The presented research results are an extension of the current state of knowledge. Detailed analyzes of the results of experimental research and a rich literature review are worth emphasizing.

The manuscript is relevant to the subject of the Materials journal and can be published in it. Before that, however, its revision is required. I have given my comments below.

  1. The tested composites were characterized by compressive strength from about 45 to 68 MPa - whether they are high-strength concretes (HSC)? Provide the criteria available in the literature on the basis of which the concrete was qualified for HSC.

  2. Keywords

    Please verify the keywords - e.g. missing name of the analyzed composite (HSC or SC)

  3. Table 1 and 2

    Please complete the units.

  4. Line 206

    A wide range of tensile strength is provided - indicate a value or range of tensile strength values for the fibers used. If the authors do not have such research results, it should be taken into account in the discussion point of the results and conclusions.

  5. Lines 274-275

    Were all the examined parameters tested on three samples? If so, please indicate clearly, and if not, please provide the number of samples on which each feature was examined. Please provide mean values and standard deviations of the examined features for each series of composites.

  6. Lines 338-340

    “almost proportional” – or also for 2% CF content?

    1. 3.2

    How was Bi-surface shear strength calculated? Please provide dependencies, because the standard procedure was not used.

  7. Fig. 15

    Please correct the drawing caption.

  8. Fig. 16

    Enter a) and b) in the appropriate picture.

  9. Line 527

    The study did not provide the density of the tested composites - please complete this.

  10. Lines 541-541

    Please provide evidence in the form of graphs, eg σ-ε, or delete the conclusion.

Author Response

(The authors gave the same response as above.)

Reviewer 3 Report

This study investigated the ductile and durable high strength concrete (HSC)
through interactive incorporation of coir waste and silica fume. The research content is very interesting and meaningful. Based on the overall quality of this study, a major revision is needed.

Comments:

(1) Introduction. More literature is needed to clarify the supplementary cementitious materials (SCMs). Use some data to highlight the importance of SCMs. Please cite the paper"Influence of waste glass powder as a supplementary cementitious material (SCM) on physical and mechanical properties of cement paste under high temperatures". This paper discussed the SCMs in the introduction and presented useful data of SCMs. “silica fume, ground steel slag, and fly ash can substantially reduce the cement content of concrete”. solid wastes from municipal, industrial and agricultural fields can be used as SCMs.

(2) Please do not use too many (Swung) dashes during references numbers such as [38-46]...

(3) Please clarify the novelty of your research in the introduction.

(4) Please provide more details of size distribution. such as mean size...

(5) Please explain how to make the distribution of fibers evenly in the concrete matrix.

(6) what is your loading mode? for shear and splitting strength. Any difference?

(7) What is the advantage of the fiber used in this study over other types of fibers?

(8) Do you get the optimum ratio of silica fume and fiber?

(9) Could you present a sustainable analysis for your concrete? To show its cost-efficient property but without compromise in performance.

(10) Please make your conclusions concise.

(11) please polish your language.

Author Response

(The authors gave the same response as above.)

Round 2

Reviewer 3 Report

The authors have addressed all my issues. Thanks for your improvement.